# Transgenerational Cycle of Traumatization and HIV Risk Exposure among Crack Users

**DOI:** 10.3390/ijerph20075285

**Published:** 2023-03-28

**Authors:** Joana Corrêa de Magalhães Narvaez, Vinícius Serafini Roglio, Brittany Di Tommaso, Flavio Pechansky

**Affiliations:** 1Department of Psychology, Universidade Federal de Ciências da Saúde, Porto Alegre 90050-170, Brazil; 2Center for Drug and Alcohol Research, Hospital de Clínicas de Porto Alegre, Universidade Federal do Rio Grande do Sul, Porto Alegre 90035-903, Brazil

**Keywords:** addiction, childhood trauma, crack cocaine, post-traumatic, gender

## Abstract

The aim of this manuscript is to understand the impact of childhood sexual abuse on the development of Post-Traumatic Stress Disorder (PTSD), Human Immunodeficiency Virus (HIV) exposure. and parental neglect in crack cocaine users, considering the role of gender. This study is a secondary database analysis of a sample from a multicenter cross-sectional study with 715 crack cocaine users receiving outpatient treatment in public mental health networks in six Brazilian capitals. Prevalence ratios were estimated by Poisson regression. In crack cocaine users with childhood sexual abuse, traumatic experiences seem to remain fixed through the development of Post Traumatic Stress Disorder (PTSD) in adulthood. Crack cocaine users with childhood abuse and PTSD in adulthood showed more sexual risk behaviors, including outcomes such as HIV (PR = 3.6 *p* < 0.001 for childhood abuse and PR = 3.7 *p* < 0.001 for PTSD). Furthermore, this traumatic trajectory affects the functional ability of crack cocaine users, especially women, to work thus impacting their inclusion and sense of social belonging. Such a chain seems to be reflected in the establishment of a circle of transgenerational transmission, to the extent that subjects with a history of abuse and PTSD reported more parental neglect towards their children. This study reinforces the importance of preventive public policies regarding early socio-emotional vulnerabilities and the need to support families, especially women, to avoid HIV and self-destructive outcomes such as crack cocaine use.

## 1. Introduction

Trauma entails a set of events, of either extraordinary or chronic exposure, and manifests in an emotional weight exceeding the individual’s ability to cope [1]. The prevalence of childhood trauma among crack cocaine users varies from 40% to 70% [2,3]. Female crack cocaine users show a high prevalence of childhood trauma, particularly of a sexual abuse nature [4,5,6]. Childhood trauma can make drugs more attractive as a form of self-medication [7,8] and as a form of anesthesia for emotional pain stemming from traumatic memories [9].

There is evidence that childhood trauma can be associated with Post-Traumatic Stress Disorder (PTSD) in crack cocaine users [3,10]. PTSD prevalence in crack cocaine users ranges from 12% to 50% [11,12]. Patients suffering from PTSD are more likely to find relief from negative symptoms through the use of cocaine. According to the self-medicating hypothesis, individuals develop dependence on drugs in order to relieve states of distress [3,7]. Therefore, those who have experienced trauma are more likely to consume higher levels of psychoactive substances and are more prone to relapse in the path of drug use, particularly with regard to cocaine [4,13,14], in order to try to eliminate the impact of their traumatic experience from their consciousness [7,15]. However, would drug use and its associated outcomes have the power to recreate more traumatization?

While previous literature has supported the notion that individuals use drugs to self-medicate past traumas or maltreatment, recent research hypothesizes about the mechanism behind the self-medication theories, since childhood maltreatment affects the development of executive functioning, with the potential to induce individual vulnerability to drug use [16].

A previous study from our group shows that the decrease in psychological function in crack cocaine users is less associated with the pattern of drug use, and more associated with childhood trauma. These subjects can easily recreate the trauma, due to the lack of neuropsychological tools, demonstrating a strong association with risk behaviors associated with drug use [3]. A study shows that female cocaine users with childhood physical neglect have significantly lower scores in cognitive flexibility, inhibitory control, selective attention, working memory, and verbal fluency in comparison to women without drug use. Additionally, lower executive functioning is associated with the occurrence of psychiatric symptoms [16]. In summary, early trauma can affect neuropsychological and emotional development [17]. Therefore, neuropsychological and emotional vulnerability in relation to problem solving can make drugs more attractive to deal with life situations.

Previous studies document the impact of many forms of trauma on substance use severity outcomes. Research shows that the severity of abuse has been associated with relapses in women. More specifically, each additional unit that females reported in the emotional abuse subscale corresponded to an increased risk of using cocaine by an average of 5% [5]. Likewise, the severity of childhood trauma, including sexual and emotional abuse, was associated with the number of days that cocaine was used by women [5]. A study on males found an interaction effect, suggesting that childhood maltreatment impairs the key mechanism of conflict resolution that is responsible for adaptive responses to stressors [18]. This demonstrates that childhood maltreatment influences the stress response, which could moderate the risk of relapse by impairing the individual’s ability to manage stress responses to drug cues [18,19]. Therefore, these individuals can be more susceptible to relapse.

Moreover, childhood sexual abuse is connected to heightened risks for behavioral, psychosocial, and physical health problems, such as HIV [20]. A study among crack cocaine users in Brazil showed an HIV seroprevalence of 11%. Although crack cocaine users were aware of their risk of HIV infection, more than half of the participants in the study reported they did not change their sexual behavior as a means to avoid risks [21]. In fact, Injecting Drug Users (IDUs) and crack cocaine users are more likely to report trading sex for drugs [22]. Furthermore, interventions that focus on trauma and psychological symptoms are an effective way to decrease HIV exposure [23]. Specifically, female crack cocaine users have been found to be in higher social vulnerability levels due to their unfortunate living conditions, lower income to support their basic needs, therefore having inadequate care, and consequently are found to have a higher prevalence of HIV [24]. Furthermore, crack cocaine users who experienced an intervention based on thought mapping and structured stories were found to increase condom use during vaginal sex by 29% compared to crack cocaine users who did not experience such interventions [25]. This suggests that HIV exposure is increased in individuals who are still experiencing stress and psychological symptoms due to childhood sexual trauma.

In summary, a great deal of research has shown that individuals who experience trauma or childhood maltreatment are more susceptible to developing crack cocaine dependence. Despite the high prevalence of childhood abuse in crack cocaine users, there are few studies about the potential impact of childhood trauma in the later development of PTSD in adulthood, as well as its outcomes to public health. There are many studies about the previous social environment among crack cocaine users, but there are no references about the impact of parental crack cocaine use on subsequent generations. The aim of this manuscript is to identify childhood sexual abuse and its impact on the development of PTSD, HIV exposure, and parental neglect in crack cocaine users, considering the role of gender. Studying early vulnerabilities can potentially help to structure more protective public policies and to stop the progression of deleterious substance use, as well as provide a clinical basis for understanding the role of early traumatization in the replication of self-destructive behaviors.

## 2. Methods

### 2.1. Study Design and Sampling

This is a secondary data analysis of a sample from multicenter cross-sectional studies with 715 crack cocaine users in outpatient treatment in the public mental health network in six Brazilian capitals (Brasília, Porto Alegre, Rio de Janeiro, Salvador, São Paulo, and Vitória) [24,26]. Data collection was conducted between April 2011 and December 2012 through interviews in the first week after treatment admission. Inclusion criteria were: (1) self-reported use of crack cocaine as the preferential drug; (2) be over 18 years of age; (3) to be in treatment for chemical dependence; and (4) to fill criteria for crack cocaine dependency according to Diagnostic and Statistical Manual of Mental Disorders (DSM-IV-TR) [27]. Subjects who did not complete the initial assessment, as well as those who had any emotional or physical impairment that limited the completion of instruments were excluded from the study.

In the subjects of this sample, crack cocaine was evaluated as a primary problem, considering a data collection that included the use of other substances, such as marijuana, sedatives, inhaled cocaine, crack cocaine /oxycodone, stimulants, hallucinogens, heroin, other opioids, and inhalants. Users were asked about age at onset, current use, years of regular use, and frequency and intensity of current use for each of these drugs.

The teams were trained under the responsibility of the regional coordinators of the six research centers, with the technical assistance and face-to-face supervision of professionals from the Center for Drug and Alcohol Research (CPAD) of the Hospital de Clínicas de Porto Alegre of the Federal University of Rio Grande do Sul (HCPA/UFRGS). In total, 6 regional coordinators and 24 data collectors were involved in the different states; regional coordinators also reviewed and performed data quality control in all steps of this study.

### 2.2. Instruments and Variables

The dataset comprises two instruments. The first is the Brazilian version of the Addiction Severity Index 6th version (ASI-6) [28,29], which consists on a multidimensional semi-structured interview that assesses the impact of substance use on the patient’s life in seven areas: medical, work, legal aspects, socio-familial aspects, psychiatric symptoms, and alcohol and other drug use. The second is the Brazilian version of the Mini International Neuropsychiatric Interview [30].

The main variables of this study were obtained from the following questions: (1) gender, accessed by question G8 of ASI-6; (2) sexual abuse and childhood sexual abuse, accessed by questions F26 (“Have you ever been sexually assaulted/abused by someone?”) and F27 (“How old were you when this first happened?”, considered to be in childhood when age < 18) of ASI-6; (3) HIV, accessed by question M9 (“Have you ever had a doctor or health professional who told you that you had HIV?”) of ASI-6; (4) Post Traumatic Stress Disorder (PTSD), assessed by MINIs diagnosis; (5) years of regular crack cocaine use, accessed by questions D27B and D28B (“For how many years of your life have you used crack in 3 or more days per week?”) of ASI-6; (6) parental neglect, accessed by questions F51 (“Have you ever been investigated or supervised by the Guardianship Council?”), F52 (“Have your children ever been removed from home by the Guardianship Council?”), F53 (“Has your parenting power ever been suspended?”), and F54 (“Are you currently responding to the custody process, or being investigated by the Guardianship Council?”) of ASI-6. Behavior is considered as parental neglect if there was at least one ‘yes’ answer to any of these questions.

### 2.3. Statistical Analysis

For the analysis, continuous data were summarized by the mean and standard deviation or the median and interquartile range, and compared between two categories by *t*-test or Mann–Whitney test according to the symmetry of the distribution. Categorical data were presented by absolute and relative frequency (prevalence) and compared using the Chi-square test of association. In the multivariable analysis, prevalence ratios (PR) were estimated by Poisson regression models with robust variance for a dependent variable with two categories [31]. Analyses were performed using the IBM SPSS software version 18 (IBM Corporation, Armonk, NY, USA) and using 95% confidence levels. Variables with a few missing data (<10) were not excluded from the analysis, so the summary tables do not match exactly with the total number of participants for all the crosstabs.

### 2.4. Ethical Aspects

This project was approved by the Research Ethics Commission of HCPA/UFRGS number 07/2014: # 14-0395, CAAE 31140014200005327. The main project and the Informed Consent Form (ICF) were replicated and submitted to the local committees of the participating centers and approved under number 100176. The evaluated individuals who agreed to participate in the study filled out individual ICFs.

## 3. Results

With regard to sociodemographic variables (summarized in Table 1), the sample consisted mostly of men (87.7%) and the overall mean age (standard deviation) was 31.5 (8.5) years. Of the sample, 40.4% declared to be white Latin American, 27.4% black, 27.7% multiracial, and 4.6% declared to be of another color and 14.1% had no schooling while 43.5% had elementary schooling, 36.9% had attended high school, and 5.5% had attended college.

Table 2 summarizes all the frequencies and bivariate analysis for the variables in the study. Gender, childhood sexual abuse, sexual abuse in life, physical abuse by someone they know, sex work, and parental neglect towards their children were all significantly associated with each other. The prevalence of all these factors were higher in women than men. Life threatening situations were more prevalent in men, in the PTSD group, and also in those who suffered physical abuse by someone known to them. The prevalence of HIV was higher in those with PTSD and also for those that suffered sexual abuse, whether in childhood or not. Among those who reported being sexually abused in childhood, the age of the first use of licit and illicit drug was earlier, and the years of regular use of crack cocaine were longer.

These intermediate analyses showed that female crack cocaine users had a higher prevalence of sexual abuse (35.3% vs. 8.2%, *p* < 0.001) and a history of childhood sexual abuse (24.5% vs. 8%, *p* < 0.001) compared to men, wherein men showed more life-threatening situations (59.2% vs. 43.1%, *p* = 0.002). Crack cocaine users with a history of sexual abuse or childhood abuse showed a greater association with PTSD (24.1% vs. 14.2%, *p* = 0.016/18.3% vs. 9.2%, *p* = 0.004). PTSD showed a greater association with HIV (45.2% vs. 13.9%, *p* < 0.001) and parental neglect (loss of custody or risk losing custody) (24.7% vs. 14.7%, *p* = 0.033). The previous experiences of sexual abuse, childhood sexual abuse, or physical abuse by someone known by them were also showed to be associated with the development of parental neglect toward their children (39.7% vs. 14.2%, *p* > 0.001/27.3% vs. 8.3%, *p* < 0.001/21.4% vs. 13.3%, *p* = 0.026). It is important to highlight that those women had higher rates of sexual abuse or physical abuse by someone known (66.7% vs. 48.4%, *p* = 0.001) and higher neglect towards their children (38.8% vs. 12.8%, *p* < 0.001). Furthermore, childhood sexual abuse was associated with physical aggression (17.2% vs. 9.2%, *p* = 0.014) and sex work (43.6% vs. 10.4%, *p* < 0.001). Furthermore, women demonstrated a higher frequency of sex work (*p* = 0.003). Childhood sexual abuse was associated with the early onset of licit and illicit drug use (in years: 13.6 vs. 15, *p* = 0.011/11.8 vs. 13.1 *p* < 0.001) and more years of regular use of crack cocaine (8.4 vs. 6.3 *p* < 0.001). Finally, subjects with PTSD had an increase in family and social problems domains (*p* < 0.001), highlighting that those women had a higher score in family and children’s problems (*p* = 0.043). Crack cocaine users with PTSD and abuse demonstrated high rates of unemployment (*p* = 0.002, *p* = 0.019), and notably women showed higher scores of problems with employment (*p* = 0.013).

With regard to the ASI’s drug use severity scores, the psychiatric score was higher among women and for those who reported PTSD, sexual abuse (either in childhood or life), and physical abuse by someone known.

Additionally, whether individuals feel that if they need help, they can count on their family members was investigated: the proportion of individuals with family support is lower in the PTSD group (75% vs. 85%, *p* = 0.015) and also in the group that reported parental neglect (70% vs. 85%, *p* = 0.002). The proportion of individuals who reported difficulties in talking about feelings or problems even with close people is higher in the PTSD group (75% vs. 57%, *p* < 0.001).

Multiple Poisson regression results are presented in a flow chart (Figure 1), considering the four variables that showed more association in the previous bivariate analysis as outcomes. We decided to include HIV as it is a major public health issue and it was significant in all analyses. The sizes of the arrows are related to the power of the association and are pointing to the outcome (dependent variable). The prevalence ratios (PR) were estimated by Poisson regression models with robust variance. Each prevalence ratio was controlled by age, educational level, and gender when these were neither predictors nor outcome, except for the following models: PTSD→CSA (childhood sexual abuse), PN (parental neglect), →CSA, and PTSD→HIV in which the severity of crack cocaine use was also added as a control; HIV→CSA, in which the severity of crack cocaine use and sex work were also added as control; CSA→HIV in which sex work was also added as a control; and Sex Work→HIV in which only the severity of crack cocaine use was used as a control. These controls were selected according to the theoretical and conceptual model and were added to improve the robustness of the analysis.

### Multiple Poisson Regressions

All the prevalence ratios presented were statistically significant and those with a narrower confidence interval provide more accurate estimates of the associations. The highest PR found refers to the increased prevalence of sexual abuse in childhood for those who practiced sex work (PR = 4.47 CI 2.78–7.20) compared to those who did not, followed by the increased prevalence of HIV among those with PTSD (PR = 3.70 CI 1.67–8.19) and childhood sexual abuse (PR = 3.59 CI 1.77–7.29). HIV prevalence was also 134% higher among those who reported parental neglect (PR = 2.34 CI 1.03–5.33) compared to those who did not. The presence of sexual abuse in childhood was more prevalent in the group with HIV (PR = 2.32 CI 1.22–4.40), in the group with PTSD (PR = 1.96 CI 1.27–3.03), and among those who reported parental neglect (PR = 2.67 CI 1.57–4.53). The prevalence of PTSD was higher in the group with HIV (PR = 2.99 CI 1.87–4.76) and among those who reported sexual abuse in childhood (PR = 1.8 CI 1.09–2.98). Finally, the proportion of women was 182% higher in the group with childhood sexual abuse (PR = 2.82 CI 1.90–4.19) than in the group without.

Years of regular crack cocaine use were employed as a proxy for the severity of crack cocaine use, therefore generating a quantitative variable that has a different interpretation from the other independent variables which are categorical. The model estimated that for each year of increased regular use, the prevalence of childhood sexual abuse (PR = 1.04 IC 1.02–1.07) and the prevalence of PTSD (PR = 1.04 IC 1.07–1.07) were both 4% higher. Assuming the linearity of this relationship in time, we can interpret that individuals with 10 years of regular use of crack cocaine present a 40% increase in the prevalence of sexual abuse in childhood and PTSD comparing to those who reported no year of regular use.

## 4. Discussion

In this study, crack cocaine users of both genders showed high rates of early sexual abuse. However, female crack cocaine users showed a higher prevalence of a history of childhood sexual abuse and sexual abuse throughout life compared to men. Users of both genders with a previous experience of childhood sexual abuse and physical abuse by someone known by them showed higher association with PTSD. Although limited by the method utilized, it is worth considering the temporal association of sexual and physical abuse in childhood and the development of PTSD in adulthood. It is possible to hypothesize that users with childhood trauma showed a non-resolutive maintenance of the traumatic experience, which remained fixed in the subject’s trajectory through higher rates of PTSD and more risk exposure for new trauma (including sexual abuse throughout life). Previous literature has already indicated this association between childhood trauma and PTSD in adulthood [10].

It is possible that sexual or physical abuse in childhood, especially in a non-protective socio-affective environment—which does not provide emotional conditions for overcoming and elaborating the situation of early violence—has impacts on the development of self-preservation functions and social skills to deal with life situations [3,17]. These early experiences of trauma have impacts on self-preservation functions, possibly driving towards a search for anesthesia of the unresolved traumatic pain [7], increasing the chances of drug use [4,10] through an earlier and severe pattern of drug use. In this sample of crack cocaine users, the previous report of childhood sexual abuse or physical abuse by someone known by them is associated with early onset use of licit and illicit drugs, a more longitudinal regular use of crack cocaine, and the overall score of the severity of drug use. Previous literature has already pointed out that childhood trauma is associated with relapse rates in cocaine users [5,18]. Besides the vulnerability to drug use, subjects showed greater vulnerability to traumatic re-exposure. It is possible that the lack of psychological tools and drug exposure can lead to more susceptibility to re-create the trauma and maintain it through continuous risky behavior exposure, which increases the development of PTSD. In addition, those crack cocaine users with PTSD have more serious patterns of drug use, seeking drugs as an attempt to numb these subsequent traumas. That cycle could be started by attempting to relieve the symptoms of an initial sexual trauma, but in fact could potentially increase through the additional opportunities for victimization [11,32] many of them experienced during the efforts made to access drugs. In parallel, the efforts undertaken to access the drugs can be potentiated in terms of risk exposure by the progression of drug use through the development of dependence and resulting in an increase in tolerance and withdrawal symptoms. The cycle continues through the attempt to soften this new trauma with drugs once again [2,33]. Indeed, the psychological dependence established by the reinforcement of drug use as the only way of emotional outflow increases the lack of emotional tools needed to face the multiplicity of life situations with the necessary dynamism and skills. A feedback loop is established, as it is known that the lack of coping behavior and emotional self-support is associated with regular crack cocaine use [4]. In this sense, subjects may be exposed to additional risks, including an impact on general health: crack cocaine users with a history of sexual abuse and PTSD showed an association with higher risk sexual behavior and HIV infection. Furthermore, crack cocaine users with a previous experience of childhood sexual abuse and physical abuse by someone they know had an association with more lifetime physical aggression episodes, showing the likehood to replicate the aggression they were early exposed to in their intimate circle. In addition, they showed more life-threatening situations, especially in men.

Our hypothesis is that a previous history of childhood sexual abuse and physical abuse, especially in the context of severe socio-environment and economic vulnerability are risk factors for the heavier use of drugs such as crack cocaine, which increases the risk of exposure to new sexual abuse, physical aggression, and higher involvement and frequency of sex work. All these factors are more prevalent among women, who are even more exposed than men, except for HIV and the number of sexual partners, which presents no distinction between genders. Many studies that investigate sex work among crack cocaine users or previous abuse do it separately, which limits the possibility of understanding the impacts of how early violence can mark the subjects’ relationship with themselves in terms of self-protection and appropriation of their bodies. A previous study shows that the meanings that women extract from past experiences of abuse and their justification for the choices made a posteriori help to explain the association between child sexual abuse and later risk mediated by sex work and crack cocaine addiction [34]. Furthermore, most studies on sexual violence, whether past or current, involve female samples of users and most of them indicate high rates of sexual abuse [4]. Yet, a mixed gender study, with a similar sample than the present research, shows that women were more likely to be in vulnerable situations, such as worse levels of education, not receiving enough money for their basic needs, more likely to be HIV positive, report sexual abuse, and be separated from their children [24], which increases crack cocaine use [4]. Studies of mixed gender samples show that sex workers initiated sex work before they first smoked crack cocaine, that 27.9% were infected with HIV [35], and also that sex-for-crack exchanges often occur in the context of intense cravings/withdrawals that may exacerbate female sex workers’ vulnerability to gender-based violence [36]. Older and more recent studies, focused internationally or in Brazil, tend to support the assertion that female crack cocaine users were significantly more likely to trade sex for drugs and/or money than men [37], to have inconsistent condom use during sexual intercourse, and reported more sexual violence [38]. In addition, the data support that women scored higher and more severely for childhood sexual abuse than men and furthermore, that a greater severity of childhood emotional abuse was associated with an increased risk of relapse in women, but not in men [5].

This continuous traumatic cycle seems to have an impact on the labor functionality of crack cocaine users, especially women, affecting their inclusion and social belonging. Traumatic experiences may have psychological and neurobiological impacts that contribute to psychopathology and psychiatric symptoms [7]. In this sense, crack cocaine users with childhood sexual abuse or lifetime sexual abuse, physical abuse by someone they know, and PTSD showed an increase in medical severity and psychiatric scores, noting that women had higher scores for both. Furthermore, crack cocaine users with PTSD and sexual abuse demonstrated high rates of unemployment, and again these were higher among women. Finally, crack cocaine users with PTSD had an increase in the family, children, and social problems domains, highlighting that those women had higher scores in family/children’s problems. A history of physical abuse by someone known increases the scores on social problems domains. Such a chain seems to be reflected in the establishment of a circle of transgenerational transmission: crack cocaine users with a history of abuse and PTSD reported more parental neglect (loss of custody or risk losing custody) towards their children. Thus, users exposed to early trauma actively replicate in their own trajectories and in their children’s destructive processes to which they were passively exposed to in their childhood.

Although it is difficult to assess the quality of parental relationships in users’ early childhood or even family coping patterns, they were exposed to trauma by people who should be protective, and that plays a role in the replication of self-destructive behaviors, maintaining the trauma in their current trajectories by more risk exposures. Some data on the current perception of their original family reaffirm the idea that users with PTSD and also those who reported parental neglect had less support from their families (with the consideration that users drug consumption should also intensify relational weaknesses and help in the breakdown of parental support). In addition, they seem to extend this pattern of limited abilities in other relationships that could potentially be places of affective sharing, as users with PTSD reported more difficulties in talking about feelings or problems. All this precariousness of psychic and social tools caused by trauma and scarce affective scenarios can potentially trigger a transgenerational component, through parental negligence in their own child. Among crack cocaine users, especially in women, this history of sexual abuse or aggression by someone belonging to their social circle may leave a mark in the development of emotional and social skills, adding an extra burden of repeating an inefficient protection net to their children alike the one they used to have (Figure 2).

In this theoretical model, all beginnings with an early trauma can be sexual or physical. However, it is not about an isolated trauma per se, but a trauma in a non-nurturing emotional context that is left unprocessed, such as physical abuse by someone known who should be protective. It is possible that this early trauma leads to a loss in the psycho-affective and neuropsychological development and self-preservation functions. It makes the subject more vulnerable to seek relief through drug use, with an early onset of licit and illicit drug use—which may also decrease neuropsychological functions and social skills—affecting the severity of crack cocaine use. During the efforts to obtain the drug (even by the progression of drug use through the development of dependence, and resulting increase in tolerance and withdrawal symptoms), this traumatized subject faces more risk exposure, more involvement in sex work, and more physical aggression episodes, showing the trend to replicate the aggression to which the subject was earlier exposed to in his/her inner circles. This creates retraumatization through new physical and sexual abuse episodes, more social and family problems, and functional damage through unemployment, all of which are more frequent among women, and with more prevalence of HIV-equally frequent in both genders. There is subsequently a new attempt to soften the new trauma through a more frequent pattern of drug use. In the end, it is not just about an earlier onset, but also about a more severe progression in the use of drugs. This leads to revictimization and creates an active relationship with trauma through the development of PTSD, affecting the subject’s functionality. Those crack cocaine users with a history of sexual abuse and PTSD showed impairment in medical and psychiatric outcomes, which are more prevalent in women. Lastly, this seems to be a transgenerational cycle, since parental neglect creates a favorable environment to restart this cycle in the next generation; again, parental neglect was higher among women. In this cycle, female crack cocaine users show an increase in many outcomes, such as early sexual trauma, sexual abuse in life, and physical abuse by someone known. Furthermore, women present more impairment of self-preservation and more opportunities to revictimization through exposure to additional sexual risk behaviors.

The present manuscript has some limitations: the first is the self-reported data collection. Despite the low credibility given to self-reports, it is a conservative limitation, insofar as those who reported the diagnosis of HIV were the ones that had access to the health care system to do the HIV test. On the other hand, we had a biased sample, since our subjects actively looked for a healthcare center, when, in fact, a naturalistic sample would be composed of subjects who did not seek treatment. Another limitation, precisely because it is a more naturalistic sample, is the use of multiple drugs, in addition to the main drug of choice, as well as the multiplicity and singularity of drug use trajectories. In this sample, users had crack cocaine as their current drug of choice, that is to say that they had made a progression from other drugs to crack cocaine. Even so, we understand that the use of multiple drugs is likely to have an impact on the user’s functioning. Since we had self-description data of drug use in the sample, the age at onset of other licit and illicit drugs as well as the regular use of cocaine were considered as factors in the analysis. Furthermore, it is important to emphasize that this study used data collected through a cross-sectional design. Therefore, it is information without precise chronology, and the results do not allow us to infer causality or interpret associations in this way. The increased prevalence found refers to the lifetime prevalence of an event being higher in one group than in the other.

The main finding that CSA was associated with female gender, HIV positivity, more extensive crack cocaine use, parental neglect, sex work, and that abuse in childhood or lifetime was associated with PTSD, replicates results from prior studies with substance-dependent adults and community populations. In a previous manuscript of our group, we identify that club users with PTSD had more history of CSA, those with previous and current symptoms of trauma showed pronounced severity in several areas, as precocity of drug use, severity of addiction. Also, those with PTSD showed greater exposure to sexual risk behavior including unprotected sex [39]. In the present study we were able to propose a much more complex, integrated, complete and authorial model which was graphically expressed to better understand the impact of exposure to early traumatization which appears to play a role in the multifactorial etiology of some health and social outcomes as substance abuse, PTSD, HIV risk exposure and child neglect replication. The organization of data culminating in a integrative theoretical model was possible due to the range of variables in the same representative sample of a continental country with social diversity and considering Multiple Poisson Regressions analyses. Additionally, the association between CSA and key domains of functioning (family, alcohol use, legal, employment, and social support) was not significant, suggesting that factors other than CSA may be more important in understanding and providing services for psychosocial impairment experienced by crack cocaine users. However, the relationship between CSA and lifetime physical and sexual abuse with sex work raises an important point that is more novel: childhood sexual or physical abuse, separately or in combination is associated with sex work. Furthermore, childhood abuse contributes to the risk of HIV positivity among crack cocaine users who do or do not engage in sex work—and PTSD mediates those relationships.

## 5. Conclusions

Crack cocaine users showed a high prevalence of a history of early sexual trauma in both genders, which is associated with PTSD in adulthood. The theoretical hypothesis for this data integration is that early abuse—in a non-supportive environment that does not allow it to be emotionally processed—may cause re-enrollment of the traumatic experience in the subject’s life trajectory. It leaves the subject vulnerable to seek relief in drug use in an anesthetic way—in order to relieve symptoms of trauma—turning into a more severe pattern of drug use. This drug use potentially recreates additional traumatization, maintaining the traumatic experience through the development of PTSD, with deprivation in self-preservation and further risk exposure, with serious outcomes for health. Moreover, it aggregates the risk of transmitting this traumatic chain for the subject’s children by parental neglect. All of these develop a retraumatization cycle, as we have termed it, a chain that involves a complex interaction: the recurrence of trauma and the resulting loss in self-preservation, added to the willingness to experience further risks, seem to be impacted by early and severe initial trauma.

In relation to crack cocaine, especially in the context of severe socio-environmental and economic vulnerability, and despite its lower prevalence, women are more exposed than men to many outcomes such as early sexual abuse. Additionally, women seem to have a more powerful impact on the transgenerational transmission of this cycle, showing higher rates of neglect towards their children. It is possible that the early sexual abuse or physical aggression by someone in her social circle may leave wounds which interfere with the development of emotional and social skills, making it harder to provide a protective net to their children which even they have had no access to. This may cause a major impact on their children’s emotional development, due to their important role in the family. This scenario ensures the important role of women in the transgenerational transmission of early traumatic exposure in the context of poor emotional support. Women also have a higher score in family and children’s problems and unemployment. It shows that the continuous traumatic cycle seems to affect their functionality and their labor capacity, having an impact on their possibilities of inclusion and social belonging.

Our data reports an important finding for the substance use field, as crack cocaine shows an increasing prevalence in developing countries such as Brazil. Data were integrated in an authorial theoretical model which graphically represented an alert to the necessity to develop preventive public policies regarding early socio-emotional vulnerabilities. There is an urgency to support families, especially women, to avoid self-destructive outcomes such as the search for anesthesia in psychoactive substances as crack cocaine. If untreated, these early traumas—and their roots—can lead to retraumatization outcomes and remain fixed in the subject’s trajectory through sequential transgenerational continuity, driving behaviors that impact public health and social markers, such as an increase in in contracting HIV and socioemotional dysfunctionalities. The increased prevalence of sexual trauma among women is therefore a social indicator with a clinical impact, which requires the development of public policies and more specific interventions.

## Figures and Tables

**Figure 1 ijerph-20-05285-f001:**
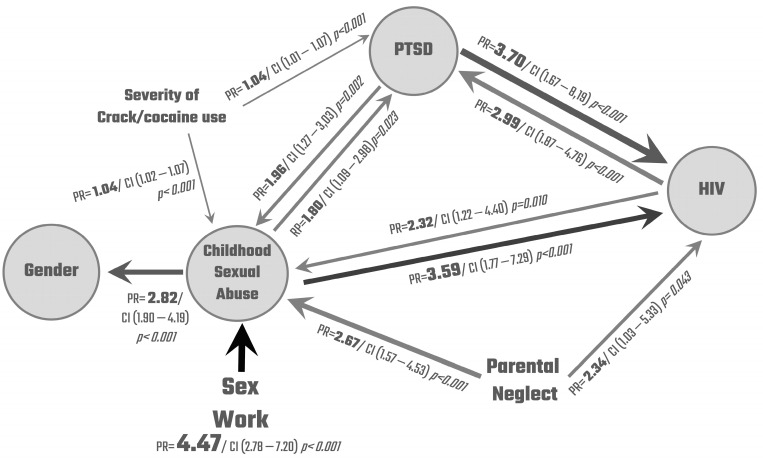
Flow chart.

**Figure 2 ijerph-20-05285-f002:**
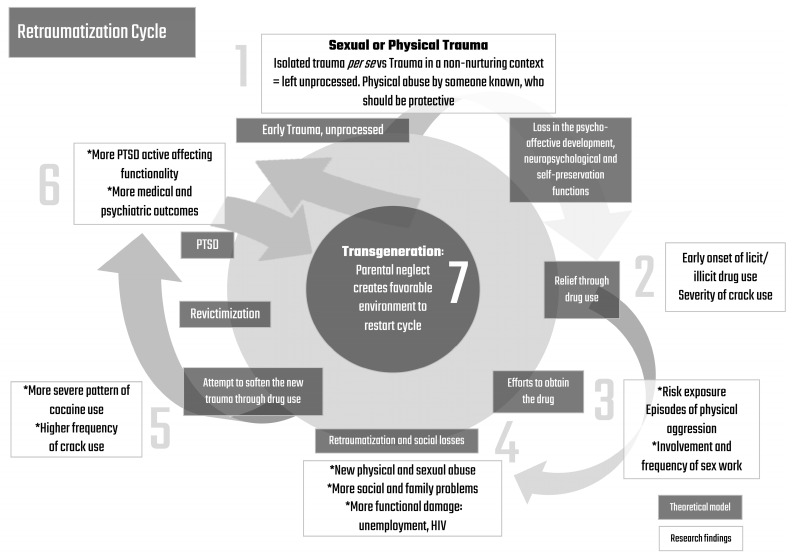
Graphic representation of the Retraumatization Cycle. * Data in which women were more exposed than men.

**Table 1 ijerph-20-05285-t001:** The sociodemographic variables of crack cocaine users in outpatient treatment units in six Brazilian capitals.

Sociodemographic Variables	
**Gender ^1,^***	
Male	611 (87.7)
Female	102 (14.3)
**Age ^2^**	31.5 ± 8.5
**Race ^1^**	
White	289 (40.4)
Black	196 (27.4)
Mixed Race	198 (27.7)
Others	32 (4.6)
**Education ^1^**	
None	101 (14.1)
Elementary School	311 (43.5)
High School	264 (36.9)
College	39 (5.5)

^1^ Absolute frequency (%). ^2^ Mean ± standard deviation. * Missing data for 2 participants.

**Table 2 ijerph-20-05285-t002:** Summary of Drug Use Patterns, HIV, Socials Outcomes, Parental Neglect and ASI-6 Severity scores by Gender, presence of PTSD, Childhood Sexual Abuse, Sexual Abuse in Life and Physical Abuse among crack-cocaine outpatients users.

	Total	Gender			PTSD			Childhood Sexual Abuse			Sexual Abuse in Life			Physical Abuse by Someone Known		
		Male	Female	*p*	Yes	No	*p*	Yes	No	*p*	Yes	No	*p*	Yes	No	*p*
Childhood Sexual Abuse ^1^	75 (10.5)	49 (8.0)	25 (24.5)	***	20 (18.3)	55 (9.2)	**	-	-	-	-	-	-	-	-	-
Sexual Abuse in Life ^1^	87 (12.2)	50 (8.2)	36 (35.3)	***	21 (19.3)	66 (11.0)	*	73 (98.6)	14 (2.2)	***	-	-	-	-	-	-
Physical Abuse by Someone Known ^1^	363 (50.8)	295 (48.4)	68 (66.7)	**	80 (73.4)	278 (46.3)	***	56 (74.7)	307 (48.0)	***	67 (77.0)	295 (47.1)	***	-	-	-
PTSD ^1^	109 (15.4)	93 (15.3)	16 (15.8)		-	-	-	20 (26.7)	89 (14.0)	**	21 (24.1)	88 (14.2)	*	80 (22.3)	29 (8.3)	***
Age of Early Onset of Licit Drugs Use ^2^	13.0 ± 3.0	13.0 ± 3.0	12.9 ± 3.1		12.9 ± 3.0	13.0 ± 3.1		11.8 ± 2.9	13.1 ± 3.0	***	11.8 ± 3.0	13.1 ± 3.0	***	12.6 ± 3.1	13.4 ± 3.0	**
Age of Early Onset of Illicit Drugs Use ^2^	15.0 ± 4.5	14.7 ± 4.2	16.3 ± 6.5		14.8 ± 4.4	15.0 ± 4.6		13.7 ± 3.5	15.1 ± 4.7	*	14.6 ± 6.0	15.0 ± 4.4		14.5 ± 4.5	15.4 ± 4.7	**
Life Threatening Situations ^1^	405 (56.7)	361 (59.2)	44 (43.1)	*	72 (66.1)	331 (55.2)	*	49 (65.3)	356 (55.7)		54 (62.1)	350 (55.9)		235 (64.7)	170 (48.4)	***
Physical Aggression ^1^	122 (17.1)	104 (17.1)	17 (16.7)		18 (16.5)	102 (17.1)		21 (28.0)	101 (15.9)	*	23 (26.4)	99 (15.9)	*	75 (20.7)	47 (13.5)	*
HIV ^1^	31 (4.3)	25 (4.1)	6 (5.9)		14 (12.9)	17 (2.8)	***	9 (12.0)	22 (3.5)	**	9 (10.3)	22 (3.5)	**	19 (5.2)	12 (3.4)	
Sex Work ^1^	39 (5.5)	25 (4.1)	14 (13.7)	***	4 (3.7)	35 (5.9)		15 (20.0)	24 (3.8)	***	17 (19.5)	22 (3.5)	***	28 (7.7)	11 (3.2)	**
Frequency of Sex Work (days in the last 6 months) ^2^	8 [2–60]	4 [2–30]	50 [7–152]	**	3 [2–5]	11 [3–70]		10 [4–90]	6 [2–60]		10 [5–40]	6 [2–60]		15 [3–65]	5 [2–60]	
Years of Regular Use of Cocaine (3+ days per week) ^2^	5.5 ± 6.9	8.6 ± 6.9	7.7 ± 7.0		10.7 ± 8.5	8.1 ± 6.5	*	9 ± 6.3	8.5 ± 7.0		8.7 ± 6.2	8.5 ± 7		8.7 ± 6.9	8.3 ± 6.9	
Years of Regular Use of Crack (3+ days per week) ^2^	6.2 ± 5.1	6.5 ± 5.2	6.5 ± 4.0		7.5 ± 5.6	6.4 ± 4.9		8.4 ± 5.5	6.3 ± 5.0	***	8.1 ± 5.4	6.3 ± 5	**	7.0 ± 5.2	6.2 ± 4.9	*
Parental Neglect Toward Their Children ^1^†	77 (17.5)	46 (12.8)	31 (38.8)	***	19 (26.4)	58 (15.9)	*	21 (41.2)	56 (14.4)	***	23 (39.7)	54 (14.2)	***	49 (21.4)	28 (13.3)	*
**ASI6 Severity Scores ^2^**																
Drugs	72.4 ± 9.7	72.4 ± 9.6	72.5 ± 10.1		72.0 ± 9.9	72.5 ± 9.6		74.7 ± 6.8	72.2 ± 9.9	*	72.8 ± 9.8	72.4 ± 9.6		73.7 ± 8.2	71.1 ± 10.8	**
Family/Child	53.6 ± 8.7	53.3 ± 8.5	55.5 ± 10.0	*	55.9 ± 10.1	53.2 ± 8.4	**	55.4 ± 10.4	53.4 ± 8.5		54.6 ± 9.8	53.4 ± 8.6		53.2 ± 8.6	54.0 ± 8.8	
Alcohol	49.9 ± 9.1	50.2 ± 9.1	48.5 ± 8.7		54.1 ± 10.0	49.2 ± 8.7	***	50.4 ± 9.9	49.9 ± 9.0		50.8 ± 10.2	49.8 ± 8.9		51.2 ± 9.5	48.6 ± 8.5	**
Psychiatric	50.1 ± 8.4	49.9 ± 8.3	51.4 ± 8.6	*	56.9 ± 7.5	48.9 ± 7.9	***	54.4 ± 8.2	49.6 ± 8.3	***	54.3 ± 8.2	49.5 ± 8.2	***	52.0 ± 8.4	48.1 ± 7.9	***
Medical	50.0 ± 9.1	49.6 ± 9.0	52.2 ± 9.4	*	53.7 ± 9.5	49.3 ± 8.9	***	53.3 ± 10.0	49.6 ± 8.9	**	53.9 ± 9.9	49.5 ± 8.9	***	51.4 ± 9.6	48.6 ± 8.4	***
Legal	51.0 ± 7.2	51.2 ± 7.3	50.0 ± 6.5		51.4 ± 7.5	50.9 ± 7.1		52.2 ± 7.8	50.9 ± 7.1		51.7 ± 7.8	50.9 ± 7.0		52.0 ± 7.7	50.0 ± 6.4	***
Employment	37.3 ± 6.3	37.1 ± 6.4	38.9 ± 5.7	**	38.7 ± 4.7	37.1 ± 6.5		38.3 ± 5.7	37.2 ± 6.4		38.6 ± 5.5	37.1 ± 6.4	*	37.3 ± 6.4	37.3 ± 6.2	
Family Social Support	47.3 ± 9.8	47.2 ± 9.9	47.8 ± 9.4		47.4 ± 9.2	47.3 ± 10.0		46.9 ± 9.6	47.4 ± 9.9		47.3 ± 9.3	47.3 ± 9.9		47.4 ± 9.6	47.2 ± 10.1	
Family Social Problem	55.2 ± 9.3	55.3 ± 9.3	54.5 ± 9.6		58.1 ± 8.4	54.7 ± 9.4	***	56.9 ± 9.2	55.0 ± 9.3		56.0 ± 9.2	55.1 ± 9.3		56.0 ± 9.5	54.4 ± 9.1	*

^1^ Summary by absolute frequency (%), Chi-Square test of association. ^2^ Summary by mean ± standard deviation or median [quartile1-quartile3], *t*-test or Mann-Whitney test. † Calculated only for the 440 participants that had children. * *p*-value < 0.05; ** *p*-value < 0.01; *** *p*-value<0.001.

## Data Availability

The data presented in this study is available upon request from the corresponding author. The data is not publicly available due to preservation of the subjects’ data privacy, in accordance with the ethical conditions of the research and the terms of written informed consent, which explicitly stated that the data would be used for a single purpose, even if comprehensive, in addition to restricting that the use was related to the project’s research team and the research groups involved in the project.

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
