# Peer review of "Transgenerational Cycle of Traumatization and HIV Risk Exposure among Crack Users"

_ijerph, 2023, doi:10.3390/ijerph20075285_

Round 1

Reviewer 1 Report

The paper constitutes substantial research. However, it would be beneficial for authors to elaborate on the research objective, indicating the added value (what does this research offer to the social welfare or to the medical foundation). A proofreading is needed so as for the text to be more profound and impeccable.

Author Response

Thank you for taking the time to carefully review our material and also for your note. In the previous manuscript version, I explored the potential clinical and public health impacts of the study only at the conclusion, after the results were already known. In this manuscript version, I added the following reflection to the objective, which also better justifies the study:

“Study early vulnerabilities can potentially help to structure more protective public policies and to stop the progression of deleterious substance use, as well as provide a clinical basis for understanding the role of early traumatization in the replication of self-destructive behaviors.”

I remain at your disposal for any further questions.

Best regards, Joana

Reviewer 2 Report

I recommend the manuscript entitled "Transgenerational cycle of traumatization and HIV among crack users" for publication in IJERPH. The paper concerns topical mental health issues.

The authors discuss the consequences of childhood experiences of sexual and physical abuse in people's adult life in a clear way. They describe the relationships between experiences of sexual and physical abuse and use of crack, PTSD and prostitution. The description of the sample, however, fails to provide information on the respondents' use of other psychoactive substances (abuse of/addiction to alcohol and abuse of other psychoactive drugs that people who use cocaine often take).

Use of other psychoactive substances apart form crack will significantly affect the respondents' cognitive, social and mental functioning.

The authors of the article argue that people who have experienced sexual abuse are likely to use crack as a way of coping with stress, trauma, negative emotions, and unpleasant memories. I believe that it is important to pay attention here to the family coping patterns which the people surveyed have developed at home and which they are currently replicating. The article also fails to emphasize the importance of psychological dependence on crack and its effects. Symptoms of the addiction will significantly affect the ways the respondents cope with stress as well as motivating their desire to repeatedly solve problems in this destructive fashion, which they view as a rewarding experience.

I recommend the paper for publication with minor revisions.

Author Response

Dear Reviewer,

Thank you for taking the time to carefully review our material and also for your note.

In relation to provide information on the respondents' use of other psychoactive substances, in the table 2, we consider age at onset of other licit and illicit drugs as one of the factors, as well as regular use of cocaine. In this sample users had crack as their current drug of choice, they had made a progression from other drugs to crack. Even so, we understand that the use of multiple drugs is likely to have an impact. We have the sample description of frequency and abstinence of each drug, but we understand that this is not the main focus of this manuscript. We agree that is a limitation, because, other psychoactive substances use will affect the users functioning. So, in this version, we add a limitation:

"Another limitation, precisely because it is a more naturalistic sample, is the use of multiple drugs, in addition to the main drug of choice, as well as the multiplicity and singularity of drug use trajectories. In this sample, users had crack as their current drug of choice, they had made a progression from other drugs to crack. Even so, we understand that the use of multiple drugs is likely to have an impact on the user´s functioning. Since we had self-description data of drug use in the sample, the age at onset of other licit and illicit drugs, as well as regular use of cocaine were considered as factors in the analysis."

Also we add on the description of sample, this sentence:

“In the subjects of this sample, crack was evaluated as a primary problem, considering a data collection that included the use of other substances, such as Marijuana, Sedatives, Inhaled Cocaine, Crack/Oxy, Stimulants, Hallucinogens, Heroin, Other Opioids and Inhalants. Users were asked about age at onset, current use, years of regular use, and frequency and intensity of current use for each of these drugs.”

In relation to the family coping patterns which the people surveyed have developed at home and which they are currently replicating.

Family coping patterns can play an important role, including triggering the re-traumatization cycle, which, in this model, is not reduced by the trauma itself, but an early trauma in an environment that does not provide affective support to elaborate the traumatic experience and that can potentially lead to a transgenerational component, through parental negligence, as we mentioned in the discussion:

“It is possible that sexual or physical abuse in childhood, especially in a non-protective socio-affective environment – which does not provide emotional conditions for overcoming and elaborating the situation of early violence - impacts on the development of self-preservation functions and social skills to deal with life situations (Back et al., 2008; Narvaez et al., 2012).”

As well in this part: “It is possible that the lack of psychological tools and drug exposure can lead to more susceptibility to re-create the trauma and maintain it through continuous risky behavior exposure”, also in this part of the text “Such chain seems to be reflected in the establishment of a circle of transgenerational transmission: crack users with history of abuse and PTSD reported more parental neglect (loss of custody or risk losing custody) towards their children. Among crack users, especially in women, this history of sexual abuse or aggression by someone belonging to their social circle may leave a mark in the development of emotional and social skills, adding the extra burden of repeating an inefficient protection net to their children, as the one they used to have (figure 2).”

Besides these notes that were already in the text, we add a sentence:

“Thus, users exposed to early trauma actively replicate in their own trajectories and in their children destructive processes to which they were passively exposed to in their childhood”.

However, we cannot speak about the relational patterns of the users' families, as we did not have access to this data. Although, because they were so exposed to early traumas, by known people, it is assumed that these exposures also play a role in the active replication of destructive behaviors and in keeping the trauma in their trajectories. Furthermore, we have access to the fact that they replicate situations of parental neglect towards their children.

But in order to discuss further this point we analyzed some additional data about the current pattern of relationship between users and their families. We add the following snippet to the results:

“Additionally, was investigated whether individuals feel that if they need help, they can count on their family members and the proportion of individuals with family support is lower in the PTSD group (75% vs 85%, p=0.015) and also in the group that reported parental neglect ( 70% vs 85%, p=0.002).

The proportion of individuals who reported difficulties in talking about feelings or problems even with close people is higher in the PTSD group (75% vs 57%, p<0.001).”

We also add in the discussion:

“Although it is difficult to assess the quality of parental relationships in users' early childhood or even family coping patterns, they were exposed to trauma by people who should be protective, and that plays a role in the replication of self-destructive behaviors, maintaining the trauma in their current trajectories by more risk exposures. Some data on the current perception of their original family support reaffirm the idea that users with PTSD and also those who reported parental neglect had less support from their families (with the consideration that users drug consumption should also intensify relational weaknesses and help in the breakdown of parental support). Also, they seem to extend this pattern of limited abilities in other relationships that could potentially be places of affective sharing, as users with PTSD reported more difficulties in talking about feelings or problems. All this precariousness of psychic and social tools caused by trauma and scarce affective scenarios can potentially trigger a transgenerational component, through parental negligence in their own child”.

 In order to reinforce the importance of psychological dependence on crack and its effects in the development of emotional tools we add the following sentences in the discussion section:

“In parallel, the efforts undertaken to access the drugs can be potentiated in terms of risk exposure by the progression of drug use through the development of dependence, and resulting in an increase in tolerance and withdrawal symptoms”.

“After all, the psychological dependence established by the reinforcement of drug use as the only way of emotional outflow, increases the lack of emotional tools to face the multiplicity of life situations with the necessary dynamism and skills. A feedback loop is established, as it is known that the lack of coping behavior and emotional self-support is associated with regular crack use (El-Bassel, 1996)”.

Furthermore, we already had the following note in the manuscript: “It makes the subject more vulnerable to seek relief through drug use, with an early onset of licit and illicit drug use - which may also decrease neuropsychological functions and social skills -– affecting the severity of crack use.

All the insertions we added in response to reviewers' notes are highlighted in yellow in the attached file.

I remain at your disposal for any further questions.

Best regards, Joana

Reviewer 3 Report

A previous study from the present authors (Narvaez and her coworkers, 2012) revealed that the decrease of psychological function in crack users is less associated with the pattern of drug use, and more associated with childhood trauma. In this thesis, the work community evaluates the results of their previous data collection with a detailed statistical analysis. The written personal data collection of about a seven hundred people was made in a group of cocaine (crack) users during or after their clinical treatment. The authors' analysis primarily studied the impact of youth violence and possible childhood sexual abuse on adult cocaine use and PTSD. The role of gender, contingent HIV infection and parental neglect in crack users have also been considered. The work required a robust mathematical and statistical analysis of the data, which, according to the reviewer, the authors performed well.

The table 2 is a bit crowded, but can be followed with some concentration. The two flow charts help to understand the recognized connections. However, the meaning of the asterisks (*) in Figure 2 is not clear (framed text boxes with numbers 3-6), this needs to be corrected.

I support the publication of the article based on my best belief.

Author Response

Dear Reviewer,

Thank you for taking the time to carefully review our material and also for your observation. In fact, we forgot to describe what the “*” in figure 2 means, now we have added the text below the table to explain (this text is highlight in yellow, in the attached file):

*Data in which women were more exposed than men.

All the insertions we added in response to reviewers' notes are highlighted in yellow in the attached file.

I remain at your disposal for any further questions.

Best regards, Joana
